# Approaches to Obtaining Water-Insoluble Fibrous Matrices from Regenerated Fibroin

**Nataliya Kildeeva** [1,*] , **Nikita Sazhnev** [1,*] , **Maria Drozdova** [2] , **Vasilina Zakharova** [1] , **Evgeniya Svidchenko** [3] , **Nikolay Surin** [3] and **Elena Markvicheva** [2]

1   Department of Chemistry and Technology of Polymer Materials and Nanocomposites, Kosygin Russian State University, 119071 Moscow, Russia; vasilinaqss@gmail.com
2   Shemyakin & Ovchinnikov Institute of Bioorganic Chemistry, Russian Academy of Sciences, 117997 Moscow, Russia; drozdovamg@gmail.com (M.D.)
3   Enikolopov Institute of Synthetic Polymeric Materials of Russian Academy of Sciences, 117393 Moscow, Russia; svidchenko@ispm.ru (E.S.)
*   Correspondence: kildeeva@mail.ru (N.K.); nsazhnev@mail.ru (N.S.)

**Abstract:** Silk fibroin (SF) holds promise for the preparation of matrices for tissue engineering and regenerative medicine or for the development of drug delivery systems. Regenerated fibroin from Bombyx mori cocoons is water-soluble and can be processed into scaffolds of various forms, such as fibrous matrices, using the electrospinning method. In the current study, we studied the correlation between concentrations of fibroin aqueous solutions and their properties, in order to obtain electrospun mats for tissue engineering. Two methods were used to prevent solubility in fibroin-based matrices: The conversion of fibroin to the β-conformation via treatment with an ethanol solution and chemical cross-linking with genipin (Gp). The interaction of Gp with SF led to the appearance of a characteristic blue color but did not lead to the gelation of solutions. To speed up the cross-linking reaction with Gp, we propose using chitosan-containing systems and modifying fibrous materials via treatment with a solution of Gp in 80% ethanol. It was shown that the composition of fibroin with chitosan contributes to an improved water resistance, reduces defective material, and leads to a decrease in the diameter of the fibers. The electrospun fiber matrices based on regenerated fibroin modified by cross-linking with genipin in water–alcohol solutions were shown to promote cell adhesion, spreading, and growth and, therefore, could hold promise for tissue engineering.

**Keywords:** silk fibroin; electrospinning; fibrous matrices; tissue engineering; chitosan; cross-linking; genipin

## 1. Introduction

Biodegradable materials based on biopolymers are widely used for various biomedical applications, namely, for the development of absorbable sutures, implants for plastic surgery, matrices for food packaging, tissue engineering, and regenerative medicine [1–6]. The creation of materials for innovative medical technologies requires the use of biopolymers and adequate approaches to their processing, which ensure their effective functioning in the human body.

Electrospinning shows promise, as it allows the fabrication of a nanofibrous matrix with a desired fiber diameter and porosity, which should be identical to native extracellular matrix (ECM) fibers and allow high adhesion rates for cell infiltration and mass transport, allowing the generation and assembly of nanofiber scaffolds with randomly oriented fiber patterns [7,8]. Additionally, the advantages of electrospun nanofibers include a small structure size, a large specific surface area, and control over morphology and composition using simple equipment [9]. Biocompatible fiber-forming polymers, most often proteins and polysaccharides, are used as spinning solutions. Sometimes, synthetic biodegradable polyesters are added to improve fiber formation [10]. Fibrous materials derived from

polysaccharides and proteins, such as chitosan and fibroin, mimic the natural habitat of the organism and thus provide optimal conditions for tissue growth and regeneration [11–15]. Chitosan substrates show promise as a matrix for tissue engineering due to their biological and physicochemical properties, as well as the presence of amino groups, which could be used for simple chemical modifications [16–19]. Silk is a high-quality natural fiber obtained from silk-spinning silkworms, mainly composed of silk fibroin and silk sericin. Of these, silk fibroin is the major component of silk, accounting for approximately 75% [20]. Its molecular chain is composed of three subunits: A heavy chain (H chain), light chain (L chain), and P25 protein, in a molar ratio of 6:6:1 [21]. The aggregated structure of silk fibroin can be classified into amorphous and crystalline regions. The highly oriented crystalline phase along the fiber axis imparts a high strength to silk. Under stress, the amorphous region of silk absorbs most of the energy, thus leading to a high toughness. Due to its impressive biocompatibility, controllable degradation, and ease of processability, silk fibroin has found applications in the fields of textiles, food, cosmetics, biomedical science, biosensing, and tissue engineering, in the form of nanofibers, microspheres, hydrogels, membranes, and scaffolds [1]. SF is a biodegradable, non-immunogenic, and nontoxic biopolymer that does not cause any side effects [1,3]. The important advantages of SF include the possibility of processing through aqueous solutions and transferring to a water-insoluble form without any chemical cross-linking reagents [22,23]. As the properties of silk fibroin continue being studied, it is increasingly being used in various biomedical fields, mainly due to its mechanical strength, elasticity, biocompatibility, and controlled biodegradability [24]. These properties are especially important in tissue engineering. A considerable number of works have been published on the processing of SF via electrospinning from solutions in various solvents, including from aqueous and mixed solutions [15,25–27]. For this purpose, regenerated SF from cocoons of the silkworm Bombyx mori is used.

The materials from regenerated SF are water-soluble and require additional hydrophobization. To prevent solubility in chitosan and improve its water resistance and mechanical properties, chemical cross-linking with non-toxic cross-linking reagents, such as genipin, can be used. However, this well-known approach is not effective enough for SF because of the rather low content of primary amino groups in this protein [28]. However, the addition of chitosan can contribute to the effectiveness of modifications with bifunctional reagents. In addition, taking into account that several possible conformational states are known for SF, namely, water-soluble $\alpha$-helices or statistical coil conformations and insoluble $\beta$-folded structures, creating conditions for the $\alpha \rightarrow \beta$ conformational transition at the modification of fibrous materials from regenerated SF could allow us to influence on the solubility of the polymer material.

Thus, in order to obtain SF-containing biopolymer matrices from aqueous solutions, in addition to the molding process itself, it is necessary to study the conditions of SF modification that lead to the loss of its solubility in water. The aim of this study was to develop an optimal method for obtaining water-insoluble fibrous matrices for tissue engineering and regenerative medicine by combining chemical cross-linking with Gp and conformational transition in SF, as well as to evaluate the biocompatibility of these matrices in vitro.

## 2. Materials and Methods

### 2.1. Chemicals

Chitosan (Mw 190 kDa, degree of deacetylation 87%) was purchased from Roeper (Germany). Raw silk in the form of cocoons and threads was purchased in China (Maks Wine). Genipin was purchased from Sigma (St. Louis, MO, USA).

### 2.2. Preparation of Regenerated Fibroin

To obtain fibroin solutions, raw silk was treated in 0.02 M sodium carbonate for 1 h to purify the fibers from contamination and to remove sericin and residual fats. Next, the silk was washed three times in distilled water, then kept in distilled water at a temperature of

90–100 °C for up to 2 h, before being left to dry at room temperature for a day. In order to optimize the conditions for the isolation of fibroin from natural silk, two methods for obtaining fibroin solutions were chosen: In an aqueous solution of LiBr and a water–alcohol solution of calcium chloride $CaCl_2:H_2O:C_2H_5OH$ (1:8:2). When using the 10% $CaCl_2$ solution in an aqueous solution containing ethanol as a fibroin solvent with treatment at 70 °C for 6 h, translucent solutions were obtained; however, during subsequent dialysis, a change in the phase state of the solution was detected, apparently associated with conformational transitions in fibroin under the influence of ethanol [29]. The optimal conditions for the separation of fibroin from natural silk were a decrease in the fibroin concentration to 8–10% and an ethanol content in the solution of no more than 30%, or the use of a 9 M aqueous solution of lithium bromide as an electrolyte [30]. These optimal conditions ensured the production of single-phase solutions suitable for the formation of films and fibers. The purified silk was dissolved in 9 M lithium bromide at a temperature of 70 °C for 1 h. The resulting solution was centrifuged and filtered to remove the insoluble conglomerates. To remove salts, it was dialyzed against water for three days with a change of medium every 3–4 h; then, it was freeze-dried and a water-soluble fibroin powder was obtained.

### 2.3. Preparation of Fibroin Solutions and Mixed Solutions of Fibroin and Chitosan

SF solutions were prepared in water using the exact weights of the dry polymer ($\pm 0.0002$ g). Dissolution was carried out on a magnetic stirrer in flasks of the required volume.

To prepare SF/chitosan mixtures with a ratio of 5:1, an appropriate amount of chitosan was dispersed in distilled water. Then, an appropriate amount of fibroin was added, and the mixture was stirred in water for 30 min. After this, an appropriate amount of glacial acetic acid was added to this mixture.

### 2.4. Kinetics of Change in Viscosity of SF Solutions

The kinetics of change in viscosity were determined using an SV-10 vibroviscometer (AND, Tokyo, Japan). After calibration, this device showed values with an accuracy of 0.1%. The viscosity change was determined automatically using the firmware at a constant temperature. The measuring range was 0.3–10,000 mPa·s. The vibration frequency of the sensor plates was 30 Hz, with the standard sample plate assuming the use of 10 mL samples. The results are presented as the average of three measurements.

### 2.5. Measurement of Turbidity

Turbidity was assessed by measuring the absorbance of the aqueous solutions of SF. Absorbance was measured at $\lambda = 400$ nm using a 10UV Spectronic Genesys (Spectronic Instruments, Inc., Rochester, NY, USA).

### 2.6. Electrical Conductivity of Fibroin Solutions

The determination of electrical conductivity was carried out on an "Expert-002" conductometer (Ekoniks, Moscow, Russia) using a submerged electrode. The test solution (5 mL) was taken and poured into the measuring cell, the electrode was immersed, and the electrical conductivity of the solution was measured 3–4 times until converging results were determined.

### 2.7. Thermogravimetric Analysis

The materials were analyzed using the dynamic method (TDM) (when the furnace temperature changed over time at a constant heating rate) on a TDM Q50 analyzer (TA Instruments) in the temperature range of 0–600 °C at a heating rate of 10 °C/min in a nitrogen atmosphere.

*2.8. Solubility of SF Fibrous Matrices*

The solubility of the SF fibrous matrices in water was studied gravimetrically, based on a weight of the SF fibrous matrix sample of 0.04 g added to water.

The mass change was calculated using the following formula:

$$\Delta m\% = (m_0 - m_t)/m_0 \times 100$$

where $m_0$ is the initial mass of the sample and $m_t$ is the mass of the dried sample after standing in water for 1 h.

The results are presented as the average of their three measurements.

*2.9. Study of Gel Formation in Fibroin Solutions When Cross-Linked with Genipin*

To determine the gelation in the systems of the chitosan/SF solution, Gp was added at different molar ratios of the cross-linking agent per chitosan amino group. The point of gelation in the system was taken as the time at which the mixture of chitosan with fibroin stopped flowing under its own weight. The results obtained are presented as the dependence of the gelation time on the ratio of the cross-linking agent—$Gp/NH_2$.

*2.10. Preparation of Fibrous Matrices by Electrospinning*

Fibrous material was obtained using the electrospinning method utilizing the free surface of a polymer solution-coated electrode in a strong electrostatic field on a Nanospider NS-Lab (Elmarco, Liberec, Czech Republic). Electrospinning was carried out based on the aqueous solutions of fibroin from the electrode surface onto the polypropylene spunbond substrate in the form of a non-woven material.

*2.11. Modification of Fiber Matrices*

Fibrous samples weighing 50 mg were filled with a solution of Gp in 80% ethanol with a Gp concentration of 0.95% and kept for 72 h, then washed with phosphate buffer solution (PBS) (pH 7.4). The concentration of the Gp solution was chosen based on previous studies [18] using the ratio $Gp/NH_2$ = 0.08 mol/mol (0.019 g Gp/g biopolymer).

*2.12. Study of Fibroin Matrix Morphology Using Confocal Laser Scanning Microscopy*

The structures of the fibrous samples were analyzed by confocal laser scanning microscopy (CLSM) using a Nikon TE-2000 inverted microscope equipped with an EZ-C1 confocal laser (Nikon, Tokyo, Japan). The fibrous samples were treated with 96% ethanol for 10 min and stained with fluorescamine. A solution of fluorescamine (0.3 mg/mL in acetone) was added to the samples, incubated at room temperature for 10 min, and then washed with saline. The excitation wavelength was 408 nm and fluorescence signals were collected at 515 ± 30 nm.

*2.13. Study of the Structures of Fibrous Matrices by Confocal Microscopy*

Images of the fiber surface and their diameters were obtained using an atomic force microscope (AFM) based on the NtegraPrima micro-console system (NT-MDT, Russia) in the cantilever semi-contact mode. The obtained data were subjected to processing and comparative analysis in the Nova SPM control program based on the INTEGRA platform and Solver.

*2.14. Cell Culture*

In the current study, a mouse fibroblast cell line (L929) and ASC52telo mesenchymal stem cells immortalized with human telomerase (hTERT) (hTERT-MSCs) were used. The L929 cell line was obtained from the Russian cell culture collection of vertebrates from the Institute of Cytology RAS, St. Petersburg. The hTERT-MSC cells were kindly provided by Prof. Efimenko from Moscow State University. L929 fibroblasts were cultured in DMEM, while hTERT-MSCs were cultured in α-MEM. Both media were supplemented

with 10% fetal bovine serum (FBS), 2 mM of L-glutamine, 1 mM of sodium pyruvate, 50 $\mu$M of 2-mercaptoethanol, 100 $\mu$g/mL of streptomycin, and 100 U/mL of penicillin. The cells were cultured in a 5% $CO_2$ humidified atmosphere at 37 °C ($CO_2$ incubator Heraeus B5060 EK/$CO_2$, Hanau, Germany).

### 2.15. Cell Cultivation in Fibrous Matrices

The fibrous samples were sterilized by incubation in 96% ethanol for 1 h. After this, the samples were washed three times with saline and incubated in 1 mL of DMEM (10% FBS) for 1 h. A cell suspension (20 $\mu$L, $10^6$ cells/mL) in the culture medium (10% FBS) was added to the fibrous samples. In 1 h, another 100 $\mu$L of medium was added to each well. Next, the plate was placed into the $CO_2$ incubator and cultivated for three days.

### 2.16. Study of Cell Proliferation in Fibrous Matrices

Cell viability was evaluated by the MTT assay. For this purpose, fibrous samples with the cells were transferred to a fresh 96-well plate, after which 100 $\mu$L of the MTT solution in DMEM (0.5 mg/mL) was added to each well, and then the plate was incubated at 37 °C for 2 h. Then formazan crystals were dissolved by adding DMSO (100 $\mu$L per well) to each well, and 50 $\mu$L aliquots were taken to measure optical density at 540/690 nm. In this study, as a control, we used cells grown on polypropylene spunbond substrate, upon which fibrous mats were formed. The relative cell viability (V) for each sample was calculated according to following equation:

$$V(\%) = \frac{OD_t}{OD_c} \times 100$$

where $OD_t$ is the optical density in the test wells and $OD_c$ is the optical density in the control wells.

### 2.17. Morphology of Cells after Three Days of Cultivation in Fibrous Matrices

After three days of cell culture on the matrices, the samples were incubated in 100 $\mu$L of culture medium (without serum) containing Calcein AM (1 $\mu$g/mL) and DAPI (10 $\mu$g/mL) for 30 min at 37 °C. The samples were then studied using a confocal laser scanning microscope (Nikon TE-2000, Tokyo, Japan) with excitation and emission at 408/515 $\pm$ 30 and 488/590 $\pm$ 50 nm for DAPI and Calcein AM, respectively.

## 3. Results

### 3.1. Effect of Silk Fibroin Solution Concentration on Fibrous Matrices from Regenerated Fibroin

It is preferable to use non-toxic solvents to obtain SF fibrous matrices for tissue engineering by electrospinning [31–33]. When using aqueous solutions, the most important factor determining the possibility of molding is the concentration of the solution [34,35]. Regenerated SF obtained using concentrated LiBr solutions, according to the procedure described in Section 2.2, takes the form of random tangles and $\alpha$-helices in aqueous solutions [27,36,37], which weakly interact with each other. The formation of the mesh required for fiber formation is possible only at rather high SF concentrations. The concentration dependences of the dynamic viscosity of fibroin aqueous solutions are shown in Figure 1.

The curve $\eta$ = f(C) has several sections with different concentration dependencies. An intensive increase in viscosity corresponding to the formation of a fluctuation grid was observed in the concentration region above 5%, which is typical for proteins. This type of dependence cannot be caused by simply an increase in the number of ionizing groups, since the electrical conductivity of the solution changes monotonously, and the pH does not exceed 0.1 units with a change in concentration of 10% (Table 1).

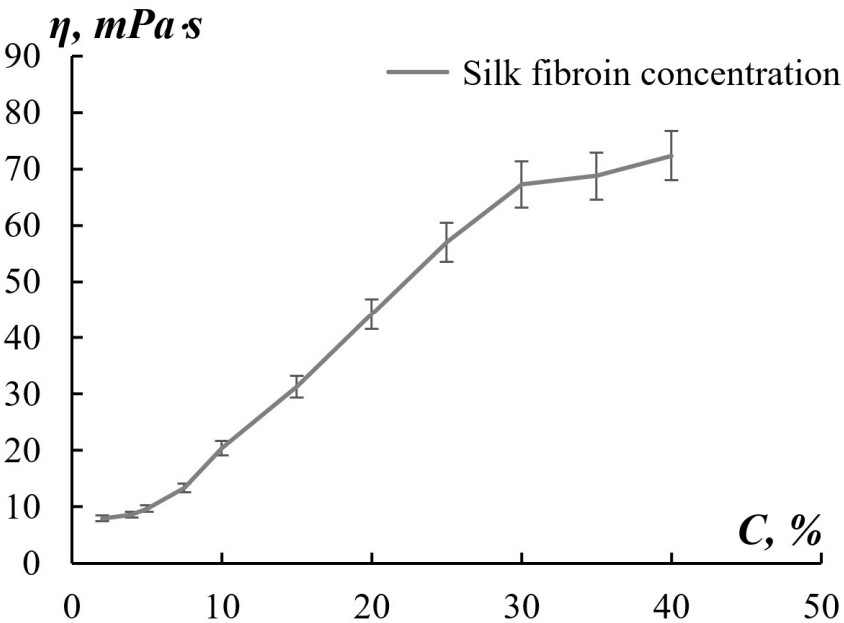

**Figure 1.** Dependence of the dynamic viscosity of the SF solution on its concentration.

**Table 1.** Properties of fibroin solutions.

| Fibroin Solution Concentration, % | pH | Conductivity $\varkappa$, mS/cm | Dynamic Viscosity $\eta$, mPa·s |
|---|---|---|---|
| 10 | 6.80 | 3.6 | 20.4 |
| 20 | 6.90 | 6.4 | 44.2 |
| 30 | 6.93 | 9.8 | 67.2 |

A sharp slowdown in viscosity growth at concentrations above 30% was found (Figure 1). This may be associated with the beginning of conformational rearrangements, namely, fibroin from the globular conformation occurring in dilute solutions transformed to the anisotropic α-helix conformation that resulted in the formation of β-folded structures. This assumption was confirmed by the results of a nephelometric study, which showed a sharp increase in the optical density of the solution in this concentration range (Figure 2).

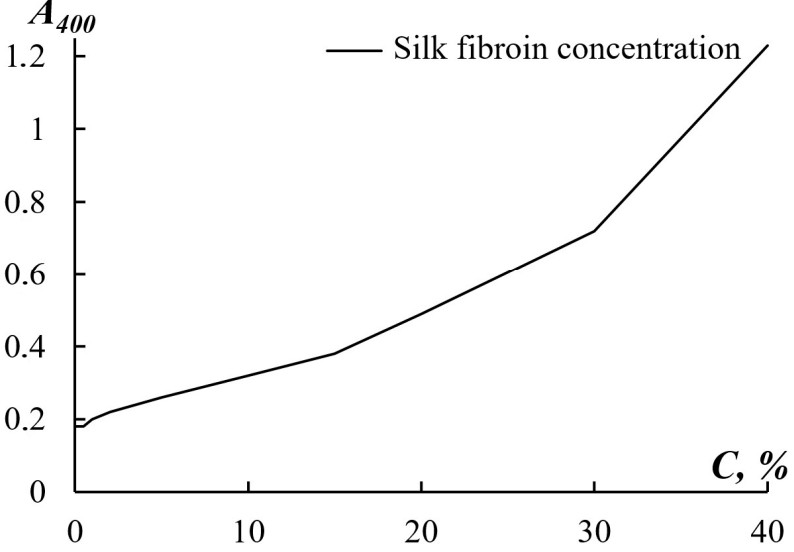

**Figure 2.** Dependence of the absorbance of the SF solution on its concentration, λ 400 nm.

Electrospinning was carried out using the free surface of an electrode coated with a polymer solution. The AFM images of electrospun fibers from SF solutions with different concentrations (from approximately 10% (*w/w*) to approximately 30% (*w/w*)) or viscosities are shown in Table 2.

**Table 2.** Effect of the aqueous fibroin solution concentration on the electrospinning process parameters.

| Fibroin Solution Concentration, % *w/w* | Electrospinning Voltage E, kV | Characteristics of the Electrospinning Process | AFM Image | Fiber Diameter, μm |
|---|---|---|---|---|
| 10 | 22.2–25.8 | Stable | | $0.61 \pm 0.22$ |
| 20 | 23.0–26.4 | Stable | | $2.80 \pm 0.20$ |
| 30 | 24.1–28.0 | Unstable | | $2.20 \pm 0.80$ |

Electrospinning and fiber formation from solutions with a concentration below 10% did not occur even at an electrospinning voltage of 30 kV, and only splashing of the solution was observed. From the AFM images, it can be seen that the SF fibers were randomly arranged and intersected with each other. The interfiber space increased with increasing fiber thickness. As shown in Table 2, at a concentration of 10%, an electrospun fibrous material consisting of fibers with an average diameter of 0.61 μm was formed. An increase in the concentration of the solution resulted in a drastic increase in the fiber diameters. Non-woven mats with a diameter of $2.80 \pm 0.2$ μm were obtained from a solution with a concentration of 20% at a voltage of 23.0–26.4 kV. The fibers had a belt-like morphology instead of the usual wire morphology. This could be due to incomplete evaporation of the low vapor pressure solvent from the thick fibers. Of an SF solution at a concentration of 30%, the morphology of the resulting material lost its pronounced fibrous structure (Table 2). The fibers stuck together, probably due to incomplete evaporation of water from a thicker jet of a viscous SF solution. The fibrous matrices obtained from a solution with a concentration of 10% were not strong enough and were destroyed during subsequent manipulations associated with the modification of SF in order to reduce solubility. The solution with a concentration of 20% (*w/w*) was selected to prepare non-woven matrices for further modification and evaluation in vitro.

### 3.2. Hydrophobization of Electrospun SF Fibers with An Ethanol Solution

The resulting electrospun SF fibers were water-soluble and required an additional hydrophobization. The conversion of fibroin to an insoluble form can be achieved by post-processing to change the random coil conformation to a more stable and water-insoluble β-sheet. The post-processing can be accomplished by treating the formed silk fibroin fibers with methanol or ethanol [38,39]. Since methanol is a toxic solvent, the formation of water-resistant fibroin materials by ethanol was used.

A change in the protein structure during the conformational transition occurs because of the redistribution of hydrogen bonds in favor of intermolecular bonds, which maintain a β-stacked conformation and impart the material resistance to water. The conformational stability of silk fibroin in an aqueous–alcoholic solution is determined by interactions of amino acid residues with each other and with a solvent. The aggregation of hydrophobic side chains is disrupted by breaking the water–alcohol hydrogen bond network and binding the alcohol to the hydrophobic group [35,36]. The formation of a β-stacked conformation is a process of rearrangement of hydrogen bonds with their formation between peptide units of different macromolecules.

Based on preliminary experiments of the solubility of electrospun SF fibers and the results described previously in [39], to treat the fibers, the material was incubated for 2 h in an 80% aqueous solution of ethanol. Structural changes in fibroin induced by the ethanol treatment were confirmed by FTIR spectroscopy (Figure 3).

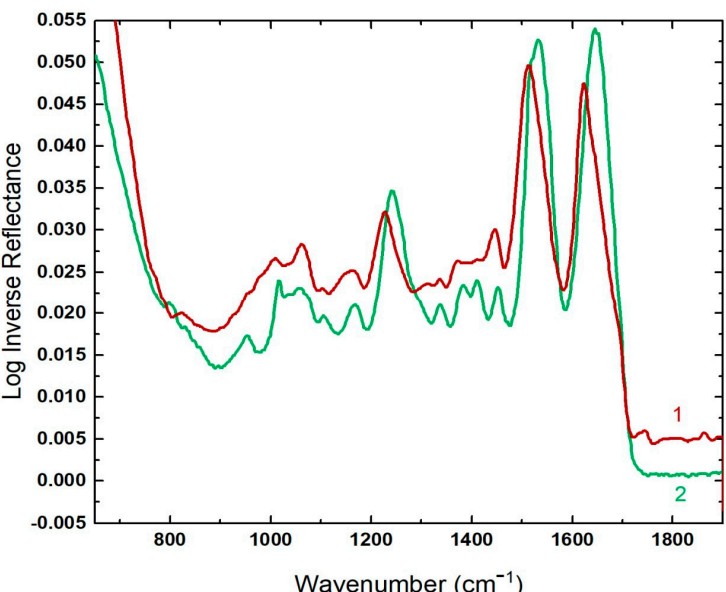

**Figure 3.** FTIR spectrum correction for internal reflectance, log 1/R, of the 80% alcohol-treated electrospun fibrous matrices from regenerated SF (**1**) and the untreated electrospun fibrous matrices (**2**).

As stated earlier, fibroin can exist in two forms. At low solute concentrations, fibroin exists as random coil/globules with an α-structure (silk I). In β-form (silk II), fibroin is ordered into antiparallel β-folded sheets formed by physical shear or exposure to solvents such as ethanol. The FTIR spectra of the electrospun fibrous fibroin mats, particularly the initial ones and those treated with an 80% ethanol solution, confirm the effect of ethanol on the conformational transformation of regenerated silk fibroin (see Figure 3). The SF fibrous materials initially had a mainly amorphous structure (silk I). The untreated sample (curve 2) showed strong bands at 1531.2 cm$^{-1}$ (amide II) and 1643.05 cm$^{-1}$ (amide I), characteristic of the α-helix and random coil conformation [36,37]. The material treated with 80% ethanol (curve 1) showed shifting bands and the appearance characteristic of the β-sheet conformation at 1513.85 and 1621.84 cm$^{-1}$ [37–40]. These changes indicate the transformation of random helices into an ordered β-sheet upon treatment with an aqueous ethanol solution. Water might even act as a swelling agent toward compact and dense silk

fibroin [41], promoting ethanol penetration and rearrangement of inter- and intramolecular hydrogen bonds. The data obtained are consistent with those described earlier [36–42].

In this work, the initial (untreated) and 80% alcohol-treated electrospun fibrous matrices were submitted to thermogravimetric analysis (TGA). This method is based on continuous recording of changes in sample weight depending on temperature under the conditions of its programmed change.

The untreated and 80% alcohol-treated electrospun fibrous matrices showed a weight loss trend with increasing temperature in the TGA plots (Figure 4). The degradation of the untreated and alcohol-treated matrices heated up to 150 °C can be attributed to the loss of bound water. In this region, the fibrous matrix sample lost its weight more slowly than the untreated sample. Water molecules bound to the β-sheet (silk II) of the fibroin structure were shown to be much stronger. The main weight loss for both the untreated and treated samples was observed in the range of 150–450 °C and was attributed to the degradation of the silk structure.

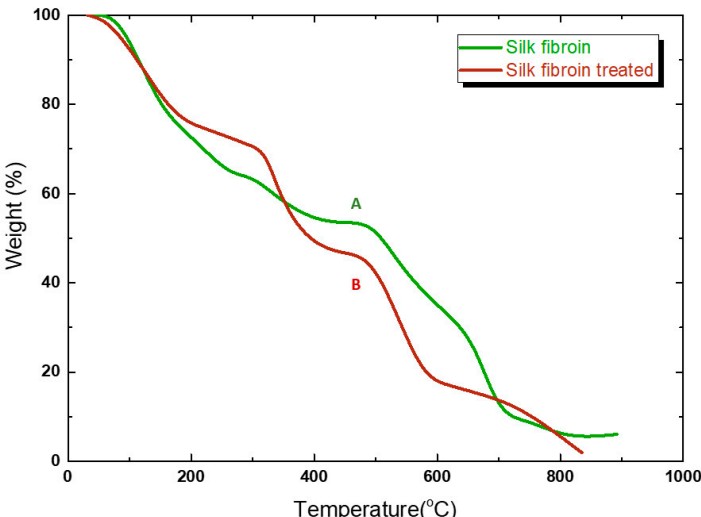

**Figure 4.** TGA plots of the untreated (**A**) and 80% alcohol-treated (**B**) electrospun fibrous matrices.

Treatment of the electrospun fibrous matrices with a water–ethanol solution caused a conditional transition of fibroin into a β-folded conformation and led to hydrophobization of the material. This was confirmed by the reduction in mass loss rate because of treatment with an 80% ethanol solution, as well as the FTIR spectroscopy results.

*3.3. Use of a Genipin Cross-Linking Reagent for Preparation of Water-Insoluble Fibrous Matrices from Regenerated Fibroin*

Another way to prevent solubility and improve water resistance and mechanical properties is chemical cross-linking with non-toxic cross-linking reagents. It is well known that chemical cross-linking of chitosan with the naturally occurring cross-linking reagent genipin causes gelation in its solutions. Upon post-processing, it leads to the production of water-insoluble fibrous, film materials, and porous cryogels [18,19]. There are different ideas about a mechanism of reaction of amino-containing biopolymers with Gp [43–46]. The interaction of SF with Gp led to an increase in viscosity, as well as resulted in an appearance of absorption in various regions of the spectrum, including the visible region (Figure 5) and blue staining of the material. Considering this and summarizing the literature data, we propose the following scheme for cross-linking fibroin with genipin (see Figure 6).

The gelation process in equiconcentrated fibroin and chitosan solutions was also investigated. Dependence of gelling time in the 2% SF solutions at pH 7.3 and in the chitosan solution at pH 5.6 during cross-linking with Gp on the cross-linking agent content is shown in Figure 7.

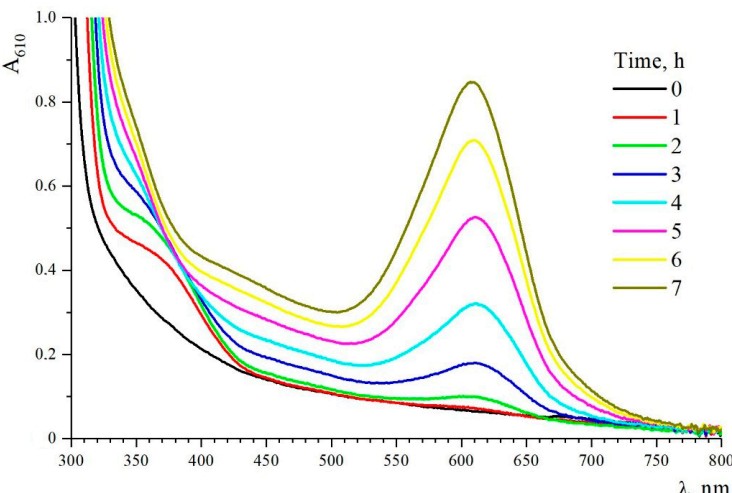

**Figure 5.** UV spectra of fibroin solutions during reaction with genipin. The fibroin concentration was 25 mg/mL, and Gp was 7.8% of the weight of fibroin.

**Figure 6.** Scheme of the fibroin cross-linking reaction with genipin.

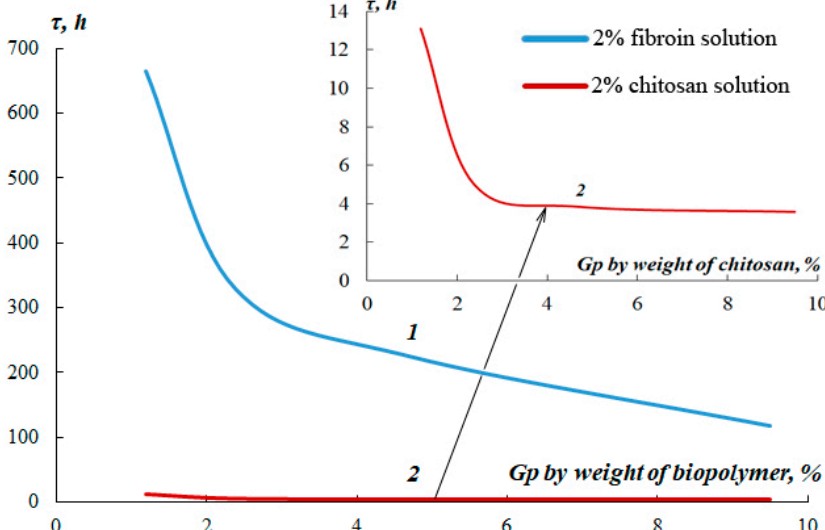

**Figure 7.** Dependence of gelling time (τ) in the 2% fibroin solution, pH 7.3 (**1**), and 2% chitosan solution, pH 5.6 (**2**), on the genipin concentration.

As can be seen from the data obtained, the gelation time in the 2% SF solution, even with a rather high Gp content and a pH value of 7.2, was 20–50-fold higher than that in the equiconcentrated chitosan solution, and it occurred for several days. This was due to a low content of primary amino groups in the fibroin molecule. Protein cross-linking using Gp occurred primarily with the participation of the amino groups of basic amino acids and, above all, lysine, whose content in regenerated fibroin was only 0.3–0.5% mol [28,47]. The terminal amino groups of the protein are not always available for modification. Therefore, it is natural that under conditions leading to chitosan gelation, the addition of Gp to an equiconcentrated solution of SF did not lead to gelation, although it caused the appearance of a blue color.

The kinetic curves obtained on a vibration viscometer are shown in Figure 8. The kinetic curves show changes in the viscosity of the SF solutions during the cross-linking of Gp at the same concentration of the cross-linking agent (10.9% by weight of fibroin) in a wide range of SF concentrations. Within 24 h, a marked change in viscosity was observed only for concentrations above 10%. During measurements for 42 h in the presence of Gp, even the 40% SF solutions did not lose their flow ability. However, the color change to green or blue (depending on the concentration of the solution), indicating a secondary reaction with the formation of a product being absorbed in the $\lambda = 600$ nm region.

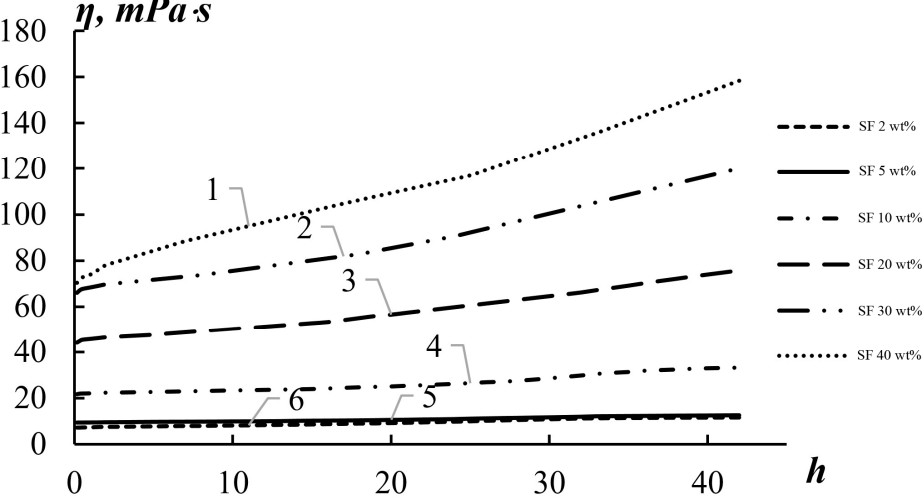

**Figure 8.** Kinetics of the viscosity change in the solutions of fibroin with genipin, pH 7.3. The fibroin concentrations were: 1—40 wt%; 2—30 wt%; 3—20 wt%; 4—10 wt%; 5—5 wt%; 6—2 wt%.

To accelerate cross-linking, leading to the formation of water-insoluble biomedical materials, we propose to use chitosan-containing systems based on SF solutions. Dehummed and regenerated fibroin is soluble throughout the pH range. Chitosan is soluble in water only in an acidic environment, when its primary amino groups are protonated and the macromolecule acquires a positive charge. Mixed solutions of chitosan and fibroin in dilute acetic acid prepared according to the procedure described in Section 2.3 were used to obtain fibrous matrices by electrospinning.

The concentration of SF in the solution was 10%. This concentration was chosen because an addition of chitosan leads to a sharp increase in viscosity, which prevents droplet formation on a free surface of the solution and the formation of extended fibers on the receiving electrode. The physicochemical characteristics of SF/chitosan solutions for electrospinning of fibrous mats are given in the Table 3. Electrospinning was carried out at a voltage of 26 kV.

**Table 3.** Composition and properties of molding compositions based on fibroin solutions.

| Concentration,% | | Fibroin/Chitosan Ratio, g/g | pH | Conductivity ϰ, mS/cm | Dynamic Viscosity η, mPa·s |
|---|---|---|---|---|---|
| Fibroin | Chitosan | | | | |
| 20 | - | 1:0 | 6.90 | 6.4 | 44.13 |
| 10 | 2 | 5:1 | 4.84 | 27.3 | 62.21 |

Genipin is soluble in both water and ethanol. Therefore, it is possible to use Gp in an 80% aqueous ethanol solution, in order to modify the electrospun fibrous matrices from SF and chitosan and to avoid their dissolution at cell cultivation. The conditions for redistribution of hydrogen bonds accompanying the formation of β-sheets were realized in an 80% aqueous ethanol solution. The presence of chitosan in the material contributed to intensification of the process of cross-linking with Gp and inclusion of the SF macromolecules in the three-dimensional grid of the cross-linked biopolymer.

The fibrous matrices were kept in a 0.95% Gp solution at 20 μL/mg for 72 h. Then they were washed with PBS (pH 7.4), which corresponded to the cross-linking degree of 0.08 mol/mol (0.019 g Gp/g biopolymer). The solubility of the resulting fibroin mats from fibroin treated with an 80% ethanol solution and a 0.95% Gp solution in 80% ethanol was studied. Modification of fibroin resulted in a decrease in solubility from 100% to 10.1%, while the introduction of genipin virtually had no effect on the sample solubility (9.6%). Modification of the electrospun fibrous matrices from SF and chitosan with a 0.95% Gp solution in 80% ethanol led to an almost complete loss of the material solubility. In particular, the difference in the weight of the untreated and treated samples dried to a constant weight was within an experimental error of ±0.002 g.

The morphology of the electrospun fibrous matrices treated with 80% ethanol and cross-linked with Gp is shown in Table 4.

**Table 4.** Morphology and biocompatibility of the electrospun fibrous matrices. Confocal laser scanning microscopy (CLSM). Scale bar = 100 μm.

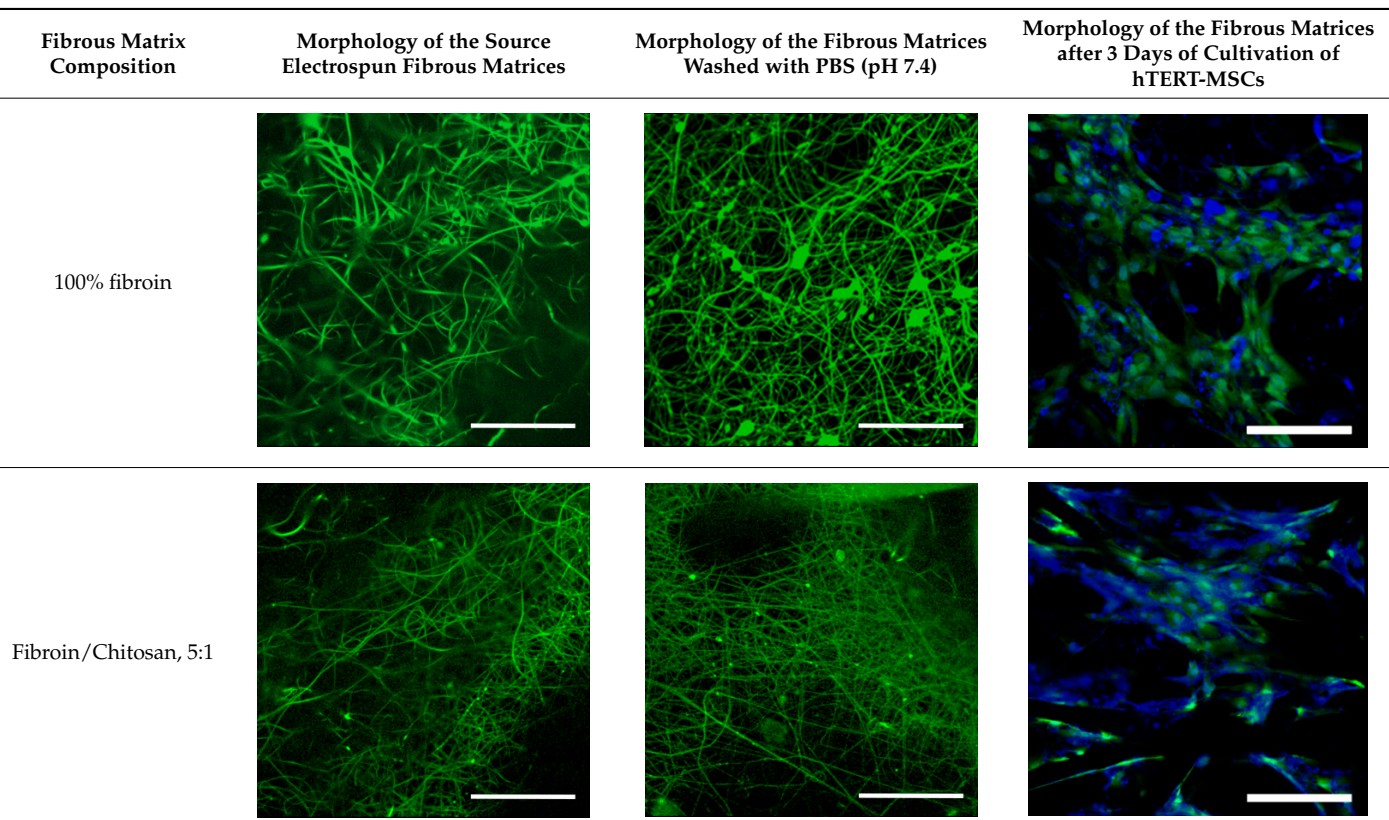

| Fibrous Matrix Composition | Morphology of the Source Electrospun Fibrous Matrices | Morphology of the Fibrous Matrices Washed with PBS (pH 7.4) | Morphology of the Fibrous Matrices after 3 Days of Cultivation of hTERT-MSCs |
|---|---|---|---|
| 100% fibroin | | | |
| Fibroin/Chitosan, 5:1 | | | |

According to the procedure described in Section 2.13, the morphology of the source electrospun fibrous matrices after being washed with PBS (pH 7.4) was studied by CLSM. As seen in Table 4, the morphology of the electrospun fibrous matrices based on 100% fibrin after being washed with PBS (pH 7.4) showed some defects, which can be explained by partial dissolution of the material, despite the fact that visually, the material retained its integrity. Therefore, hydrophobization of the SF fibers after their treatment with Gp in an 80% ethanol solution was insufficient. This likely occurred because of a rather low concentration of Gp amino groups available for cross-linking in SF. An introduction of chitosan, which, in the process of cross-linking with Gp, promoted the inclusion of SF within a three-dimensional network of cross-linked biopolymers, improved the water resistance, which resulted in a reduction in matrix defects.

Polysaccharide and protein-derived fibrous materials can mimic ECMs and are widely used in tissue engineering. To evaluate the biocompatibility of the fibrous matrices, immortalized mesenchymal stem cells (hTERT-MSCs) were seeded on the fibrous matrices and cultured for three days. Before cell seeding, the fibrous samples were sterilized with 96% ethanol for 1 h. The cell morphology and distribution were monitored using transmission light and confocal microscopy. In order to assess cell viability, the cells were stained with a vital dye, Calcein AM, and DAPI after three days of cultivation (see Table 4). As seen in the CLSM images, the cells penetrated the fibrous matrices and adhered and spread on the fibers. Moreover, the conversion of non-fluorescent Calcein AM to the green fluorescent dye Calcein, which occurs only in living cells, indicates that these green-stained hTERT-MSCs were still alive after three days of cell culture within the fibrous matrices. Moreover, L929 mouse fibroblasts were cultured in the matrices for three days and cell viability was estimated using an MTT test. We did not reveal any differences in cell growth and proliferation. Thus, the cell viability was 103% and 106% in case of matrices from 100% fibroin and fibroin/chitosan 5:1, respectively. Cells grown on polypropylene spunbond substrate were used as a control (100%).

## 4. Conclusions

Fibrous matrices from SF were obtained from water solutions with concentrations of 10% and 20% using the electrospinning method. Materials from regenerated SF are soluble in water and require additional hydrophobization. Two methods to prevent solubility of fibroin-based matrices were studied: Conversion of fibroin to β-conformation by treatment with an ethanol solution and chemical cross-linking with genipin. The TGA method revealed stronger binding of water to the material after its treatment with an 80% ethanol solution. This, along with the results of FTIR spectroscopy, indicates that treatment with aqueous–ethanol solution electrospun fibrous matrices causes the conformational transition of fibroin into β-folded conformation and leads to hydrophobization of material. The interaction of Gp with SF leads to the appearance of a characteristic blue color, but does not call for the gelation of solutions. To speed up the cross-linking reaction with Gp, it is proposed to use chitosan-containing systems and modify fibrous materials via treatment with a solution of Gp in 80% ethanol. An approach combining chemical cross-linking with the natural cross-linking reagent genipin and the $\alpha \rightarrow \beta$ conformational transition in fibroin is proposed. Thus, a method for preparing water-insoluble fibrous matrices for use in regenerative medicine and tissue engineering was optimized. Electrospun fiber matrices based on regenerated SF, modified by cross-linking with Gp in water–alcohol solutions, support cell growth and proliferation and may be promising for tissue engineering.

**Author Contributions:** Conceptualization, N.K. and E.M.; methodology, N.S. (Nikita Sazhnev); software, N.S. (Nikita Sazhnev); validation, M.D., V.Z., and E.S.; formal analysis, V.Z.; investigation, N.S. (Nikolay Surin); resources, N.S. (Nikita Sazhnev); data curation, N.S. (Nikolay Surin); writing—original draft preparation, N.S. (Nikita Sazhnev), V.Z. and M.D.; writing—review and editing, N.K.; visualization, N.S. (Nikita Sazhnev); supervision, N.S. (Nikolay Surin); project administration, N.K.; funding acquisition, E.M. All authors have read and agreed to the published version of the manuscript.

**Funding:** This research was funded by the Russian Science Foundation (project No. 22-13-00261).

**Institutional Review Board Statement:** Not applicable.

**Informed Consent Statement:** Not applicable.

**Data Availability Statement:** All of the data are available within the manuscript.

**Acknowledgments:** The Center of Collective Use, Enikolopov Institute of Synthetic Polymeric Materials of Russian Academy of Sciences, Moscow, Russia.

**Conflicts of Interest:** The authors declare no conflict of interest.

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
