# Peer review of "Approaches to Obtaining Water-Insoluble Fibrous Matrices from Regenerated Fibroin"

_technologies, doi:10.3390/technologies11050146_

Round 1

Reviewer 1 Report

Kildeeva et al. fabricated silk fibroin nanofiber matrices, and further explored their properties for potential tissue engineering application. Some concerns and issues should be addressed before publication.

1. Abstract must be presented in a better and clear way. The background is too long. The authors paid more attention on what they did. Instead, some more descriptions about results and conclusions should be introduced. Moreover, some important result data should be presented in this section.

2. Please state the reasons why silk fibroin was employed. What are the merits and demerits of silk fibroin compared with other natural biopolymers like collagen?

3. Some more descriptions should be added to introduce the recent advances of electrospinning-based scaffolds in Introduction section. Some related works (10.3390/nano13071150, and10.3390/technologies11020048) should be discussed.

4. In the chemicals section, some important information were missing. For instance, the company for cocoon purchasing. please state the reasons on the selection of parameters for the exaction of silk fibroin. Were any preliminary experiments conducted?

5. How did the author make sure the weight ration of SF and chitosan? The authors should give the detailed testing parameters for each testing method.

6. How many replicated were selected for the testing of dynamic viscosity? The SD value should be added in Figure 1.

7. The characteristics of different electrospinning samples may need a bit more discussions. For example, how about the mechanical properties of different samples?

Moderate editing of English language is required.

Reviewer 2 Report

At present so many backgrounds in abstract. One sentence is enough.

Abstract has many paragraphs it should be one.

Hours should be written as h. Make a space between oC and number.

Decimal should be written as point (.). For ex. Table 1, 0,61 should be written as 0.61. check all these errors.

Fig. 3 need to redraw in scientific software. Another figure also. Don’t directly paste from instrument software. It is not acceptable.

How authors know the crosslinking and how much did authors calculate. Please include it. Authors can follow this and cite it in the text accordingly. Progress in Organic Coatings 174, (2023) 107305

Did the authors test solubility test?

Insert these references in the text, Food Packaging and Shelf Life 33 (2022) 100904; and https://doi.org/10.1016/j.lwt.2012.11.018. Insert more recent studies. Preferably 2023 and 2022.

At present conclusion is so long please include the main findings.

What is roughened data from AFM images.

Table 4, authors should present cell viability test in order to present biocompatibility studies through MMT or CCK-8 assay.

Rheological study needs to be present for the gelation test.

Need to improve better way.

Round 2

Reviewer 1 Report

The reviewer's comments have been addressed.

Reviewer 2 Report

The authors improved the manuscript. 

English is fine.